# Examining Evidence of Benefits and Risks for Pasteurizing Donor Breastmilk

**Margaret E. Coleman** [1,*]**, D. Warner North** [2]**, Rodney R. Dietert** [3] **and Michele M. Stephenson** [4]

[1] Coleman Scientific Consulting, Groton, NY 13073, USA
[2] NorthWorks, San Francisco, CA 94133, USA; northworks@mindspring.com
[3] Department of Microbiology and Immunology, Cornell University, Ithaca, NY 14850, USA; rrd1@cornell.edu
[4] Syracuse University, Advancement and External Affairs, Syracuse, NY 13244, USA; mmstephe@syr.edu
* Correspondence: peg@colemanscientific.org

**Abstract:** An evidence map is visualized as a starting point for deliberations by trans-disciplinary stakeholders, including microbiologists with interests in the evidence and its influence on health and safety. Available evidence for microbial benefits and risks of the breastmilk ecosystem was structured as an evidence map using established risk analysis methodology. The evidence map based on the published literature and reports included the evidence basis, pro- and contra-arguments with supporting and attenuating evidence, supplemental studies on mechanisms, overall conclusions, and remaining uncertainties. The evidence basis for raw breastmilk included one benefit–risk assessment, systematic review, and systematic review/meta-analysis, and two cohort studies. The evidence basis for benefits was clear, convincing, and conclusive, with supplemental studies on plausible mechanisms attributable to biologically active raw breastmilk. Limited evidence was available to assess microbial risks associated with raw breastmilk and pasteurized donor milk. The evidence map provides transparent communication of the 'state-of-the-science' and uncertainties for microbial benefits and risks associated with the breastmilk microbiota to assist in deeper deliberations of the evidence with decision makers and stakeholders. The long-term aims of the evidence map are to foster deliberation, motivate additional research and analysis, and inform future evidence-based policies about pasteurizing donor breastmilk.

**Keywords:** breastmilk microbiota; benefit–risk; colonization resistance

## 1. Introduction

Profound bioactivities of raw breastmilk include benefits attributed to the dense and diverse natural microbiota, termed natural microbiota hereafter [1–5]. Benefits appear to decrease significantly with feeding of pasteurized donor milk and infant formula in hospital neonatal intensive care units (NICUs) around the world [6–8].

Infants benefit from breastfeeding not just nutritionally, but by both 'seeding and feeding' the infant gut [9–12], providing microbes that seed the naïve gastrointestinal (GI or gut) ecosystem and nutritive components that feed both infant and microbial cells. Many studies in the past decade have characterized both the natural microbiota of mammalian milks, including human breastmilk, and the benefits that the natural microbiota of milks provide to developing gut, immune, respiratory, and neural systems [3,11]. The 'microbial seeding and feeding' of the natural microbiota of milks in mammalian gut systems is now understood to contribute to 'completeness' of the natural microbiota of the gut essential to both stimulating balanced development of the immune system and providing 'optimized colonization resistance', suppression of enteropathogen growth and infection, with enhancement of clearance of enteropathogens, by the natural microbiota of healthy breastfed infants' gastrointestinal systems [9].

Preterm and low birthweight infants appear to suffer higher risk of failure to thrive and morbidity and mortality from infections than full term infants, as discussed more fully

in Sections 3 and 4. One of the leading causes of infant mortality, severe inflammatory necrotizing enterocolitis (NEC), appears to be a disease of dysbiosis rather than associated with infection by a specific enteropathogen [13,14]. Thus, a uniquely vulnerable niche may exist in the dysbiotic gut ecosystem of preterm infants who lack a protective natural microbiota of the gut and may ingest pasteurized donor milk, lacking or depleted of the natural microbiota of the mother's own milk or raw breastmilk from donors.

Formal methodologies for microbial risk assessment and benefit–risk assessment are ideal for developing evidence-based policies for pasteurizing human donor milk. However, no benefit–risk assessments or risk assessments for bacterial pathogens were identified that compared pasteurized and raw milks from humans. Regarding global risk management for donor breastmilk, Japan and Norway [15,16] choose to provide raw donor breastmilk to NICU infants. All human donor milk banks in Norway (See https://europeanmilkbanking.com/country/norway/, accessed on 29 December 2020) and some in Germany (See https://europeanmilkbanking.com/country/germany/, accessed on 29 December 2020) screen and provide raw donor breastmilk, as documented on the European Milk Bank Association website.

Fear and dread of microbes as germs that will kill us may factor strongly into a policy that is becoming more controversial with expanding knowledge of the natural microbiota of milks from -omics studies in this decade: the decision to require pasteurization of breastmilk from donors in most human milk banks around the world [17–21]. The fear of microbes as germs may entrench well-meaning scientists and regulators in misconceptions of 20th century science, and wall them off from full consideration of the tremendous advances in knowledge about the natural microbiota of milks, particularly the rich body of evidence for both benefits and risks of raw breastmilk. In addition, germophobia may be fueled by misinformation about formula milk to families around the world that discourages breastfeeding, despite the significant loss of benefits associated with infant formula compared to breastfeeding [22]. Regarding risk management for breastmilk and formula, a companion manuscript submitted to this special collection [23] addresses relevant issues of 'managing our microbes' and raises concerns about potential tradeoffs between economic motivations of the global infant formula industry and health benefits of raw breastmilk to infants and families.

No formal quantitative benefit–risk assessment or risk assessment was identified to date that assessed raw breastmilk and pasteurized donor breastmilk or the conditions when risks of pasteurized donor breastmilk might outweigh the benefits of raw breastmilk with its natural microbiota intact.

For this study, formal methods for benefit–risk assessment [24–26] were considered as potential frameworks for future communications with diverse stakeholders about the 'state of the science' and uncertainties for evidence of benefits and risks to NICU infants fed raw breastmilk and pasteurized donor milk. Qualitative methods were selected for structuring recent evidence related to benefits and risks of raw and pasteurized breastmilk as evidence maps [25] in order to provide a transparent and comprehensive characterization of the extent and nature of the body of evidence for deliberation and open public discourse. Consistent with Wiedemann and colleagues [25], we also view the evidence map as intended for engagement of all interested stakeholders and decision makers, including microbiologists, but not necessarily for the general public.

Work on an evidence map for the breastmilk ecosystem was undertaken to provide an accessible visual format for beginning broad societal deliberations about the complex body of evidence and the current state of knowledge for benefits, risks, and uncertainties associated with the breastmilk ecosystem. The qualitative evidence map for the natural microbiota of the breastmilk ecosystem is envisioned as a starting point for future exercises of analysis and deliberation planned for this project, rather than as a quantitative analysis of benefits and risks that might be undertaken in the future.

## 2. Methodologic Approach

The extensive literature on the natural microbiota of human milks and evidence on benefits and risks was compiled over multiple years of this project (January 2017 to July 2021), including results of searches, as well as manuscripts and reports provided by scientists and regulators. Combinations of key words (e.g., raw OR unpasteurized, milk, microbiota OR microbiome, benefit, risk, human OR breastmilk OR breastfeeding) were used in searches of PubMed, Google Scholar, and the Cochrane database of systematic reviews in combination with discipline-specific key words by multiple authors. Additional studies were identified by forward searching studies citing key references by discipline. This study focused on recent reviews and human experimental and observational studies. A comprehensive working bibliography for the project is available upon request.

The recent evidence was structured using an evidence map, as described by Wiedemann and colleagues [25], for the raw breastmilk ecosystem (Figure 1). The evidence map includes a blue text box for a 'Pro-Argument' about benefits and a 'Contra-Argument' about risks. Human experimental evidence linking infants fed with breastmilk to clinical disease or health outcomes from cohort studies, randomized controlled clinical trials, and other observational studies, as well as systematic reviews, meta-analyses, and quantitative microbial risk assessments or QMRAs, would be introduced under the center section associated with the 'Pro-' and 'Contra-Arguments'. Supplemental studies on plausible mechanisms for benefits and risks of the breastmilk microbiota for the 'Pro-' and 'Contra-Arguments' are listed in green beveled-edge boxes on the left section of the evidence map. See Supplementary Table S1 for brief summary information on the relevance of each study for future deliberations of benefits and risks. On the right section of the evidence map is a brief summary section about the evidence basis, conclusions for benefit and risk arguments, and remaining uncertainties.

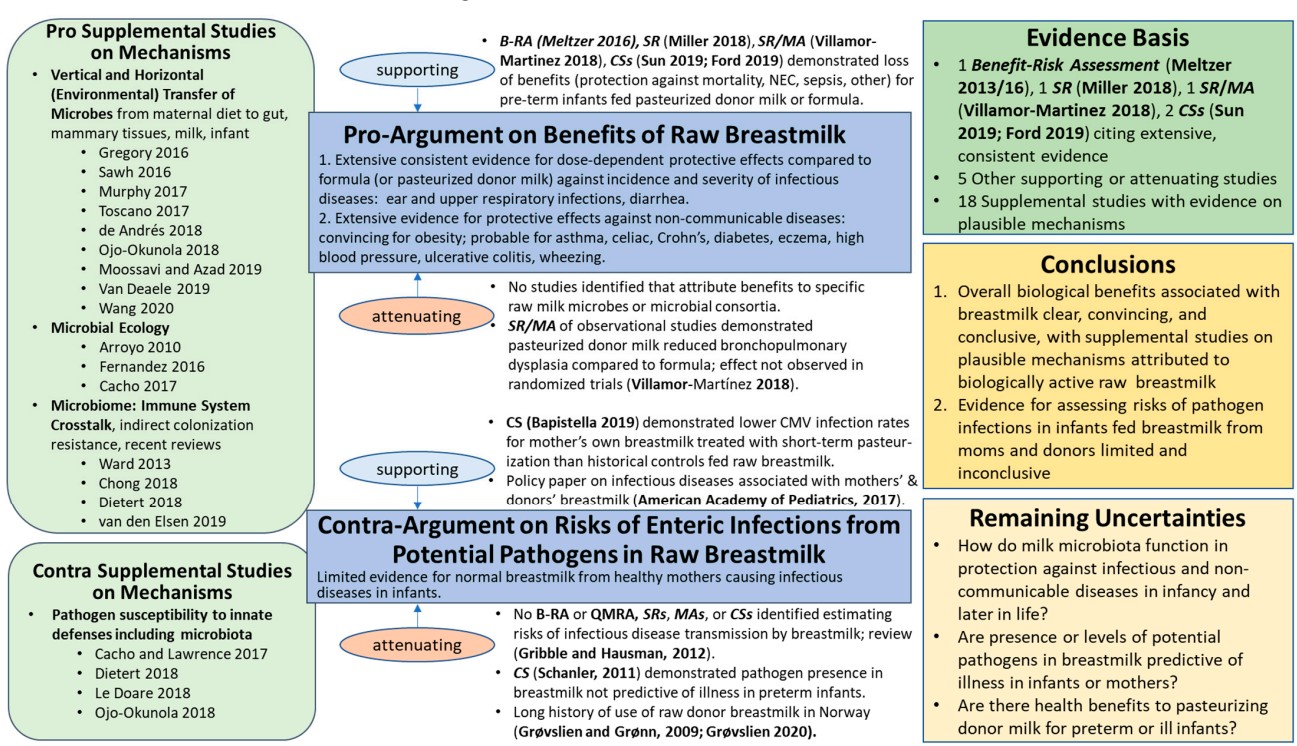

**Figure 1.** Evidence Map for Raw Breastmilk Ecosystem. B-RA = Benefit–Risk Assessment; QMRA = Quantitative Microbial Risk Assessment; CS = human cohort study; MA = meta-analysis; R = review; RT = randomized trial; SR = systematic review; and NEC = necrotizing enterocolitis. Referencing within this figure lists first author and date for those references cited in the text and subsequently with reference numbers: Meltzer et al., 2016 [27]; Meltzer et al., 2013 [28]; Miller et al., 2018 [6]; Villamor-Martinez et al., 2018 [7]; Sun et al., 2019 [29]; Ford et al., 2019 [8]; Bapistella et al., 2019 [30]; American Academy of Pediatrics, 2017 [17]; Gribble and Hausman, 2012 [31]; Schanler et al., 2011 [32]; Grøvslien and Grønn, 2009 [15]; Grøvslien, 2020 [33]. For Supplemental Studies on Mechanisms, first or first and second author(s) and year are listed within the figure, and full references provided in Supplementary Materials.

The features of evidence maps emphasized by Wiedemann (the evidence base, the pro- and contra-arguments, conclusions, and remaining uncertainties) were applied to the body of evidence for raw and pasteurized milks, with supplemental studies informing mechanisms of the arguments mapped separately.

## 3. Results

The evidence map for the raw breastmilk ecosystem (Figure 1) includes a pro-argument on the benefits of raw milk and a contra-argument on the risks of raw milk. For each argument, highlights of systematic reviews, meta-analyses, randomized trials, cohort studies, and reviews are provided which support or attenuate each argument. Supplemental studies that provide evidence of potential mechanisms for benefits and risks are introduced in Figure 1 (and Supplementary Table S1).

Due to the high diversity of the natural microbiota of milks in mammals and the variability of conditions influencing its composition and abundance [2–5], ambiguity of results and conflicting studies are to be expected. Thus, the body of evidence presented in this section is a starting point for future deliberations, research, analysis, and development of evidence-based policies about pasteurizing donor breastmilk proposed in Section 5.

### 3.1. Raw Breastmilk Ecosystem

Extensive, consistent evidence of benefits was identified from recent systematic reviews (one including a meta-analysis), a subsequent cohort study, and a benefit–risk analysis for the pro-argument on the benefits of raw breastmilk. In contrast, limited evidence was identified for the contra-argument on the risks of enteric infections from potential pathogens in raw breastmilk.

### 3.1.1. Benefits: Pro-Argument

This analysis focused on two recent systematic reviews/meta-analyses [6,7] and a benefit–risk assessment of breastmilk for infant health in Norway [28], summarizing an earlier report [27]. Miller [6] noted considerable overlap with the studies considered by Villamor-Martínez [7]. The three studies applied and cited standard methods for assessing quality of the evidence and presenting pooled results. We refer readers with interest in the original research to these studies.

Miller et al. [6] considered clinical trials and observational studies published between 1990 and 2017. These focused on the benefits of breastmilk in reducing morbidity and mortality in neonates, including the primary cause of mortality in NICU infants, a severe inflammatory disorder of the intestine of pre-term infants termed necrotizing enterocolitis (NEC). Other adverse health endpoints documented for neonates and children included late onset sepsis (LOS), retinopathy of prematurity (ROP), bronchopulmonary dysplasia (BPD), and neurodevelopment in infants born ≤28 weeks gestation, and/or studies on infants of mean birth weight of ≤1500 g. Additional information on these primary diseases of infants is provided below.

Miller identified 49 studies, of which 44 were included in their meta-analysis. Six studies were randomized trials (representing a total of 1472 infants), and 43 observational studies (representing a total of 14,950 infants). Risk of bias was judged low for all randomized trials, and low (26), moderate (14), and high (3) for observational studies. The authors noted that many studies had low sample size and were not designed or powered to detect small differences.

The authors concluded that breastmilk provided: (i) a clear protective effect against NEC (~4% reduction in incidence and 2% reduction in severity); and (ii) a possible reduction in late onset sepsis (LOS), severe retinopathy of prematurity (ROP), ROP, and severe NEC. Evidence was judged inconclusive regarding potential benefits to neurodevelopment. The authors noted that any volume of breastmilk is better than exclusive pre-term formula, and the higher the breastmilk dose, the greater the protective benefit to infants. The authors considered the evidence for benefits for pasteurized donor breastmilk inconclusive.

Villamor-Martínez and colleagues [7] considered randomized controlled trials and observational studies (cohort and case–control) published before 2017 that compared pasteurized donor milk with pre-term formula and raw breastmilk (mother's own milk) in reducing BPD in very preterm neonates (<32 weeks gestational age) or very low birth weight (<1500 g) infants. Other adverse health endpoints documented were days of mechanical ventilation and days on oxygen.

Villamor-Martínez included 18 studies (7 randomized and 11 observational) in their analyses. The authors noted a significant reduction in bronchopulmonary dysplasia (BPD) risk for raw breastmilk compared to pasteurized donor milk from two random trials. The authors also noted that raw breastmilk is more effective in reducing risk of multiple morbidities than pasteurized donor milk.

The findings of the Villamor-Martínez study from observational studies comparing raw breastmilk with pasteurized donor milk groups and pasteurized donor milk vs. preterm formula groups are discussed in the Attenuating section below.

Meltzer et al. [28] presents evidence and analysis on benefits of breastmilk against possible risks from exposure to contaminants in breastmilk, focusing on conditions relevant in Norway. The benefit assessment was generally based on systematic reviews and meta-analyses published between 2003 and 2013. Although the authors conducted a full literature review for possible risks associated with breastmilk, the risk assessment portion of the study is less relevant to infectious and non-infectious disease effects consistent with the focus of the current study.

Three outcomes characterized by Meltzer and colleagues have direct relevance to the current study: (i) convincing evidence of beneficial effects of breastmilk on neurodevelopment; (ii) convincing evidence for a protective effect of breastmilk on infections (acute otitis, and gastrointestinal and respiratory infections), immune response-associated diseases (celiac disease, Crohn's disease, inflammatory bowel disease, and ulcerative colitis), and type 1 diabetes; and (iii) convincing evidence of a protective effect of breastmilk on growth abnormalities (overweight and obesity in childhood). Overall, the study concludes that:

> 'the benefits of breastmilk clearly outweigh the possible risks from contaminants' for neurodevelopment, defense against infections, and growth abnormalities. Probable beneficial effects later in life were noted for type 1 and type 2 diabetes, as well as high blood pressure. No conclusions were drawn regarding other diseases of the immune system (allergy, asthma and wheezing, atopic dermatitis) due to 'inconclusive results on the benefit side and few and disperse studies on the risk side'.

Supporting

Two additional cohort studies [8,29] published after the two systematic reviews [6,7] are summarized here in support of the pro-argument on benefits of raw breastmilk.

Ford and colleagues [8] conducted a prospective cohort study to compare gut microbiota and health outcomes of 74 preterm infants (<1500 g birth weight) fed mothers' own milk (raw breastmilk) with 43 similar infants fed pasteurized donor milk in the NICU at Texas Children's Hospital. Significant increases associated with raw breastmilk were measured: diversity of the infant gut microbiota ($p < 0.001$; abundance of protective genera *Bifidobacterium* ($p = 0.02$) and *Bacteroides* ($p = 0.04$)), whereas pasteurized donor milk was associated with increased abundance of potentially pathogenic *Staphylococcus* ($p = 0.02$). Raw breastmilk was associated with a 60% reduction in feeding intolerance ($p = 0.03$ by multivariate analysis) and greater growth ($p < 0.01$, weight gain; $p = 0.03$ length; $p = 0.02$ head circumference; $p < 0.01$, growth velocity). Pasteurized donor milk was associated with higher composite outcome of severe morbidity (necrotizing enterocolitis (or NEC), spontaneous intestinal perforation (or SIP), sepsis, severe BPD, or death; $p = 0.02$ adjusted).

Sun and colleagues [29] conducted a multi-center prospective cohort study to address the feasibility and safety of providing fresh mothers' own milk within 4 h of expression (raw breastmilk) at least once daily for a study duration of 32 weeks. The study excluded a

formula feeding intervention. Sun enrolled 207 very preterm infants at NICUs in China (221 mother–infant pairs), and all infants were initially provided intravenous feeding (total parenteral nutrition). Oral feeds were introduced as soon as possible after birth using a nasogastric tube. Fourteen of the mothers recruited for the intervention group were unable to produce sufficient breastmilk to continue in the study.

Once full enteral feeding was possible for both groups, fortifiers and other nutritional supplements including probiotics were added and, subsequently, vitamins and iron were added. The intervention group (n = 98 infants) was fed raw breastmilk at least once daily for 32 weeks, and 109 control infants were fed exclusively defrosted human milk from donors or mothers (all frozen). Donor milk was reported only pasteurized if cytomegalovirus (CMV) density in donor milk exceeded $10^4$ copies per mL. Infants were moved from enteral feeds to breastfeeding when clinically advised. The authors noted that 10 preterm infants (three in the intervention treatment, seven in the control group) died before discharged from the NICU.

The outcomes for the study included feasibility (measured as percentage of mothers providing daily breastmilk sample and other metrics) and metrics for morbidity and mortality (infant growth; mortality; sepsis; NEC; ROP; BPD; intraventricular hemorrhage; and a post hoc metric, composite outcomes of NEC and mortality).

Statistically significant decreases in adverse health effects ($p < 0.05$) were noted for the intervention group for the following metrics: ROP; BPD; sepsis; and the combination of NEC and mortality. Note that although the authors reported that mortality alone was similar between intervention and control groups, the trend of lower mortality for the intervention group was significant at $p < 0.10$. Additionally, the duration of mechanical ventilation and total parenteral nutrition were lower for the intervention group. The authors concluded that feeding raw breastmilk once daily to preterm infants in the NICU was 'safe, feasible, and may reduce morbidity'. The study raised the concern that practices in NICUs and human milk banks (freezing, pasteurization) deprives infants of benefits of the cellular content of natural breastmilk, predominantly bacterial and immune cells killed by thermal challenges [29].

Attenuating

No studies were identified that attributed health benefits to specific raw milk microbes or microbial consortia, independent of other bioactive factors present in raw milks.

Two subsets of the data considered in the systematic review/meta-analysis of Villamor-Martínez and colleagues [7] provided some evidence attenuating the pro-argument for benefits of raw breastmilk. Analysis of data from eight observational studies (including either human or bovine fortifier) yielded high overall significance for a protective effect against BPD due to pasteurized donor milk compared to preterm formula. Analysis of data from four observational studies did not find significant overall differences between raw breastmilk and pasteurized donor milk in prevention of BPD, although the largest of these studies did demonstrate protection for the raw breastmilk group at $p = 0.07$. Thus, some evidence supports benefits for pasteurized donor milk over formula, apparently associated with heat-stable components of breastmilk. However, the studies did not compare raw and pasteurized breastmilk directly.

Supplemental information on plausible mechanisms for benefits of breastmilk microbiota summarized below were noted by Ojo-Okunola and colleagues [34], with the suggestion that these mechanisms operate optimally in synergy rather than as discrete independent mechanisms, in humans and other mammals. Further information from supplemental studies listed in Figure 1 are briefly tabulated in Supplemental Table S1. These plausible mechanisms include the following five examples:

1. Vertical transmission of microbes from mother to infant, including in utero microbes, anaerobes typical of the GI tract, strains of oral probiotic supplements during pregnancy and lactation, and microbiota of breast tissue, vaginal tissues, and skin;

2.    Anti-infective functions, including colonization resistance by commensals for resistance to acute infections and induction of oral tolerance;
3.    Immunomodulatory activities of T-regulatory cells, peripheral blood mononuclear cell subsets, cytotoxic T-cells (CD8+), natural killer (NK) cells (non-specific), and cytokines for milk microbiota. Decrease exaggerated inflammatory responses to colonizing bacteria (commensals and opportunistic pathogens under certain conditions);
4.    Anti-allergic properties attributed to LABs and commensals of milk microbiota that decrease the occurrence and severity of allergic responses in animal models and some human studies; and
5.    Metabolic activities of LABs and commensals essential for digestion of oligosaccharides into short chain fatty acids (SCFAs) that become an energy source for host cells in the colon, thus increasing nutrient availability and absorption for the host. Further, Sozańska [35] cited studies demonstrating that SCFAs in the GI tract enhance the epithelial barrier function of the gut, influence bone marrow dendritic cell maturation, and inhibited Th2-dependent response, interconnecting metabolic functions to functions 2, 3, and 4 above.

In addition to the five plausible mechanisms noted above, anti-tumor properties have been observed [34], but this class of activity is not discussed further herein.

### 3.1.2. Risks: Contra-Argument

No systematic reviews, cohort studies, or quantitative microbial risk assessments (QMRAs) were identified that estimated the likelihood of infectious diseases transmitted to infants from raw breastmilk.

### Supporting

A recent policy statement by the American Academy of Pediatrics (AAP) [17] included some evidence supporting the contra-argument for raw breastmilk risks. This report [17] cited studies documenting microbial contamination [36,37] in breastmilk donated via milk banks. Keim and colleagues [36] also compared prevalence and levels of microbial contamination between breastmilk purchased from from the Internet (101 samples) and breastmilk from milk banks (20 samples).

However, these studies also include some evidence attenuating the contra-argument under certain conditions. In spite of potential microbial contamination of raw breastmilk, Keim and colleagues [36] stated that milk banks typically screen donors or donor milk for some viral diseases. In their view, the benefits of feeding unpasteurized mother's own breastmilk to hospitalized infants 'likely outweigh the risk of bacterial disease'. The AAP [17] also acknowledged that breastfeeding (mother's own milk) during hospitalization is always preferred and should be encouraged, citing five studies demonstrating loss of cells, macronutrients, anti-inflammatory factors, and potential probiotic organisms with pasteurization.

The AAP recommends only pasteurized donor milk collected from screened donors and distributed through established human milk banks. The AAP recommends against use of unpasteurized donor milk from direct, internet-based, or informal milk sharing.

The AAP cited the Human Milk Banking Association of North America (HMBANA) with overseeing nonprofit human milk banks in the US and Canada. Most HMBANA milk banks use Holder pasteurization (heating at 62.5 °C for 30 min) to process raw donor milk.

The key points developed by the AAP [17] include preference for feeding mothers' own milk (raw breastmilk) when available and pasteurized donor milk for preterm or ill infants whose mothers are unable to provide sufficient raw breastmilk. Other key points include screening of donors, pasteurization and post-pasteurization testing, discouragement of use of internet donor milk or raw breastmilk sharing, and access to pasteurized donor milk to appropriate high-risk infants based on medical necessity, regardless of an individual's financial status or ability to pay.

A subsequent cohort study conducted at NICUs in Germany [30] documented significantly decreased rates of CMV in very preterm infants whose mothers were seropositive for CMV. Mother's own breastmilk treated with 'short-term pasteurization' (heating at 62 °C for 5 s; 87 infants) was associated with CMV infection in 2 of 87 (2.3%) infants in the treatment group (recruited from 2010–2012) compared to 17 of 83 (20.5%) infants in the historical control group fed untreated (raw) mother's breastmilk (retrospective cohort from 1995–1998).

Attenuating

Three studies include evidence attenuating the contra-argument for raw breastmilk risks. One is a brief review by Gribble and Hausman [31] that pointed out the paucity of scientific evidence supporting the theoretical transmission of infectious disease to infants via raw breastmilk. Of viruses examined to date, cytomegalovirus (CMV), human immunodeficiency virus (HIV), and human T-cell leukemia virus (HTLV) appear to be transmitted via breastmilk, but the latter two appear to require repeated exposures over a long period of time to actually cause infection. Further, the authors state [31] that while a majority of mothers are infected with CMV, the presence of CMV in breastmilk is only a problem for premature infants. Cohort studies were cited that estimated the rate of disease transmission at 0.6 to 4% of infants breastfed for 6 months by HIV-positive mothers. Bacterial pathogens (*Streptococcus*, *Salmonella*, or *Listeria*) apparently rarely cause infant disease via breastmilk, and other potential bacterial pathogens that could be present in milk (*Klebsiella*, *Pseudomonas*, *Staphylococcus*, or *Bacillus*) have not been demonstrated to cause disease in infants via breastmilk.

The second attenuating study [32] enrolled a cohort of 161 mothers and their 209 infants to test the hypothesis that the presence of microbes in expressed breastmilk was correlated with subsequent infections in preterm infants (<30 weeks post gestational age).

Milk samples (n = 813) were collected weekly and evaluated for presence and levels of Gram-positive and Gram-negative bacteria and presumptively identified using standard culture methods in the Manual of Clinical Microbiology published by the American Society for Microbiology. All isolates of potential pathogens from infant samples of blood, cerebrospinal fluid, or urine were identified. Relative risks were estimated from initial milk culture positives that appeared as a pathogen in the paired preterm infant(s).

Bacterial isolates in milk were classified first by Gram-stain reactions. Approximately 49% of milk samples had low densities of Gram-positives ($<10^4$ cfu/mL) and 5% high density Gram-positives ($\geq 10^4$ cfu/mL). Low density ($<10^3$ cfu/mL) Gram-negatives were not detected, and 0.4% of milk samples had high density Gram-negatives ($\geq 10^3$ cfu/mL). In addition, results for mixed Gram-positive and -negative milk samples were reported (17.5% low density Gram-positives and -negatives; 14.3% low Gram-positives and high Gram-negatives; 7% high density Gram-positives and -negatives; and 2.5% high density Gram-positives and low-density Gram-negatives).

Of 1963 isolates from milk, the predominant species (present at $\geq 5\%$ total isolates) are summarized in Table 1, collectively accounting for 87.3% of isolates. (Note that the pathogenic potential of the isolated strains was apparently not tested using in vitro or in vivo models or for sequence homology with known virulence genes.)

**Table 1.** Predominant microbial isolates from mechanically expressed breastmilk derived from Schanler et al. [32], Table 2, p. 336).

| Predominant Bacteria Isolated from Breastmilk | Number of Isolates | Percent of Isolates |
|---|---|---|
| *Staphylococcus epidermis* | 548 | 42.4 |
| *Enterococcus faecalis* | 152 | 11.8 |
| *Acintobacter* spp. | 127 | 9.8 |
| *Streptococcus* spp. (coagulase-negative) | 89 | 6.9 |
| *Stenotrophomonas maltophila* | 79 | 6.1 |
| *Enterobacter* spp. | 69 | 5.3 |
| *Staphylococcus aureus* | 65 | 5 |

The authors reported that bacterial isolates identified in the initial milk samples were sporadic and not predictive of subsequent milk sample positives. Further, there were no significant relationships between total or specific bacteria and densities and NEC, surgery for NEC, duration of antibiotic use, time to full feeding, or duration of hospitalization. The odds of infection in infants before or after exposure to breastmilk containing potential pathogens were not significant for Gram-positive (*Staphylococcus*, *Streptococcus*) and most Gram-negative bacteria (*E. coli*, *Enterobacter*, and *Klebsiella*). The distributions of bacteria in milk did not match bacteria isolated from pathology samples (infant blood, cerebrospinal fluid, and urine).

Schanler and colleagues [32] questioned the utility of screening breastmilk by traditional culture methods for predicting infectious potential. The authors and other cited studies observed that exposure to potential pathogen in breastmilk appears to cause adverse effects (infections) in few infants. Further, the authors suggest that co-occurrence of potential pathogens in breastmilk and infants may reflect common nosocomial exposures to mother and infant rather than transmission of potential pathogens from breastmilk.

The third study attenuating the contra-argument for raw breastmilk risks [15] described a long tradition for milk banks in Norway to provide raw donor breastmilk, with strict controls and rigorous screening of donors and trace-back if needed for investigation of adverse events as practiced by blood banks.

Norwegian milk banking has continued since 1941, with rigorous donor screening comparable to screening of blood donors. Routine use of raw donor milk was reportedly based on screening of donors, a low incidence of CMV and hepatitis B and C via blood testing of donors, and the high standard of living in Norway. In addition, donors are screened for human T-cell leukemia virus (HTLV 1 and 2). Approved donors negative for these agents had not become positive upon regular re-testing, as of 2009. Norway continues to provide raw donor breastmilk as the best choice for premature or ill infants whose own mother's milk production is insufficient.

The first author of this study provided updates on guidelines and practices used in Norway for screening donor breastmilk [33]. Each container of donor milk (250–500 mL, received frozen) is screened for microbes (total counts and pathogens). In 2019, 698 of 4317 L of donor milk (16%) was discarded due to the presence of potential pathogens, rather than pasteurizing donor milk for feeding to preterm or full-term infants [33].

### 3.1.3. Benefit–Risk Conclusions

Overall biological benefits associated with breastmilk are clear, convincing, and conclusive, with supplemental studies on plausible mechanisms attributed to biologically active raw breastmilk based on the body of evidence summarized herein (Figure 1, Supplemental Table S1a).

Evidence for assessing risks of pathogen infections in infants fed raw breastmilk from healthy mothers and donors is limited based on the lack of benefit–risk assessments or QMRAs that include data on potential pathogen frequencies and levels in raw breastmilk, health effects in exposed infants, and simulations of effectiveness of alternative risk management scenarios (e.g., screening and pasteurizing donor breastmilk) intended to reduce risks to infants. Nor were such data available in systematic reviews, meta-analyses, or cohort studies. Some supplemental studies on plausible mechanisms for benefits and risks are noted herein (Figure 1, Supplemental Table S1b).

### 3.1.4. Remaining Uncertainties

Although plausible evidence exists for multiple mechanisms likely to contribute to health benefits associated with raw breastmilk, further research is needed to deepen understanding of the dynamics and functionality of components of the raw breastmilk ecosystem so that benefits to at-risk children, particularly preterm, malnourished, and stunted infants and children around the world [38], can be maximized.

Three questions frame future research needs for the raw breastmilk ecosystem.

1.  How do milk microbiota function in protection against infectious and non-communicable diseases in infancy and later in life?
2.  Are presence or levels of potential pathogens in breastmilk predictive of illness in infants or mothers?
3.  Are there health benefits to alternative pasteurization conditions for feeding donor milk to preterm or sick infants? Particularly, what risk management strategies (e.g., conventional Holder pasteurization and alternative conditions such as short-term pasteurization) provide optimal balance of health benefits and protection against neonatal disease?

## 4. Discussion

Although the state of knowledge about the microbiota of milks has expanded tremendously in the past decade, no benefit–risk assessment or microbial risk assessment was identified to support current policies for pasteurizing human donor milk at milk banks in the US and many other countries around the world. Evidence and analysis documenting the frequency and levels of potential pathogens in raw breastmilk are needed for estimating risk to neonates. One study documents that the presence of potential pathogens in breastmilk was not predictive of illness in infants [32], likely due to the protective effects of the natural microbiota of breastmilk. However, regulators around the world appear to continue to rely, not on the body of evidence, but on the 'precautionary principle' [39]. The biological complexities of the breastmilk ecosystem merit deeper and more transparent analysis and wider deliberations to effectively apply the current state of the science and the state of uncertainties for re-evaluating benefits and risks attributable to the breastmilk microbiota.

Certainly, substantial time, energy, and resources would be required to compile and assess the complex microbial evidence published even in the most recent decade as relates to policy decisions about pasteurization. A recent study [40] documenting effectiveness of anti-viral activity for various thermal treatments of donor milk spiked with CMV (temperatures from 48 °C to 63 °C, times from 1 to 30 min) recommended that current policies specifying conventional Holder pasteurization be re-evaluated. It is unclear at present what risk management strategies might provide an optimal balance between maximizing benefits and minimizing risks to neonates. However, profound beneficial bioactivities of raw milks, including the natural microbiota, appear to decrease significantly for multiple disease endpoints and multiple studies compared to pasteurized breastmilk [6–8,29].

Wiedemann and colleagues [25] described the 'link' between two opposing world views as 'information'. These authors also described evidence maps designed as structured argumentation as a path for communicating information to bridge opposing world views and increase reliability and transparency for resolving misunderstandings and mixed messages that invoke fear and dread.

The opposing world views about donor breastmilk considered herein could be described as follows. One world view is that raw breastmilk is inherently dangerous. Such claims invoke fear and dread, yet rarely reference relevant peer-reviewed studies. The opposing worldview is that raw breastmilk benefits infants, even at-risk infants. The evidence map (Figure 1) documents the 'state of the science' and uncertainties. Thus, application of this qualitative methodology for evidence mapping represents unique and relevant analysis to open deliberations between people with opposing world views. Our long-term aims are to increase transparency about the available evidence for the controversial issues around milk pasteurization, foster deliberation, motivate research and analysis, and inform future evidence-based policies about pasteurizing donor breastmilk.

Our work on the evidence map for the breastmilk ecosystem (Figure 1) is consistent with evidence that without raw breastmilk to support growth and development of a healthy natural microbiota in the gut, infectious disease and abnormal development of the immune system may be more likely, potentially contributing to inflammatory diseases, asthma, and allergies later in life [41,42]. Health of preterm and ill infants in the NICU environment may also be dependent on regular ingestion of the natural microbiota of raw breastmilk

for proper maturation and differentiation of the immune, neural, and respiratory systems early in life [9–12].

Pasteurized donor milk benefits and risks are of grave concern for preterm or ill infants in NICUs that typically provide only pasteurized human donor breastmilk or formula [20,43], both lacking the raw milk microbiota and other key bioactive components, when mother's own milk is insufficient or unavailable. At present, the pasteurization policies of human milk banks appear inconsistent with the available evidence in the 21st century, including the 'state of the science' and uncertainties illustrated in the evidence map (Figure 1). The extensive body of evidence from human cohort studies conducted around the world demonstrates significant decreases in health benefits to neonates fed pasteurized donor milk when compared with fresh raw breastmilk [6–8,27–29]. Donor milk banks appear to require pasteurization based on the possibility that potential pathogens may be present in raw breastmilk without rigorous assessment of the body of evidence. However, due to the lack of formal benefit–risk or QMRA studies that include the body of evidence on the natural microbiota of breastmilk, it is uncertain if the loss of health benefits associated with feeding pasteurized donor milk in the NICU is outweighed by the theoretical risks of acute infectious illness from raw donor breastmilk or mother's own milk.

Layered effects of the breastmilk microbiota and metabolites appear to exert beneficial effects, both acute and long-term, on inflammation, immune development, and immune-mediated diseases [41,42,44]. Neonates who are fed pasteurized donor milk or infant formula lacking a protective microbiota appear to assume measurable health risks (depressed growth, greater risk of NEC, sepsis, and mortality) and loss of measurable benefits observed in infants fed raw breastmilk complete with its natural microbiota. Risks associated with pasteurized donor milks may exceed benefits to NICU infants.

### 4.1. Might Scientific Revolutions Be Shifting Our Paradigms?

Dietert and Silbergeld [45] pointed out the need to insert the microbiota into our frameworks for assessing safety and risk to human superorganisms, holobionts of *Homo sapiens*, and microbial partners in health [46]. The evidence map for the breastmilk ecosystem generated herein (Figure 1) illustrates the major shifts in methods, concepts, and knowledge base [39] that microbiologist and physician Martin Blaser [47] described as the 'microbiome revolution'. The advances of the 'microbiome revolution' are defining a 'new normal science' regarding the structures and functions of the dense and diverse microbes in our bodies and in milks as our partners in health, a new paradigm diverging from the 20th century paradigm of microbes as germs that could kill us.

Consistent with Thomas Kuhn's book, *The Structure of Scientific Revolutions* [48], this 'change in paradigm' (fundamental change in basic concepts and experimental practices, theories, models, or patterns of a scientific discipline) for raw breastmilk is supported by an extensive body of evidence on benefits and risks of human milk that, from our perspective, strongly challenges the validity of many outdated societal notions about interactions between the microbiota of milks and the host systems, particularly the immune system. For example, the notion that the presence of potential pathogens in raw milks certainly pose too high a health risk to permit even healthy infants, as well as pre-term or sick infants, to consume raw milks appears invalid. A paradigm shift may be necessary to incorporate advancing knowledge of milk ecosystems to develop policies that appropriately balance the benefits and risks of the raw milk microbiota for the 21st century.

Brief perspectives are offered below, highlighting key studies that illustrate the urgency for updating the 'state of the science' and uncertainties in order to begin new journeys on the difficult terrain of a shifting paradigm for considering the evidence for benefits and risks of pasteurizing donor breastmilk. Without a paradigm shift, development of evidence-based decision support for raw breastmilk and pasteurized donor milks seems impossible. Key updates are offered from 21st century perspectives of the natural microbiota of the

breastmilk ecosystem, particularly microbiologic cross-talk with cells of innate and adaptive immune systems, and decision science disciplines.

### 4.2. Updating Earlier Notions from Science, Medicine, and Risk Analysis

The body of evidence for factors influencing the microbiota of milks related to the 'environment' aspect of the traditional 'disease triangle' (e.g., air quality and pollution; diet; supplements and pharmaceuticals; behavior/lifestyle/environment including farm and non-farm environments, built and natural environments, dust, soil; and water) is extensive and relevant to modeling dose–response relationships for pathogens amidst the natural microbiota of raw breastmilk and GI systems [49]. Some notable recent studies relevant for future dialogue with stakeholders include the following [50–52].

The 20th century notion that the microbiota of milks are simply contaminants posing high risk to human health appears invalid. Perceptions of bacteria as germs to be eradicated are gradually being replaced by deeper awareness of symbiotic (commensal and mutualistic) microbiota as our partners in health [9,46,53].

As humans are now recognized as a majority-microbial superorganism rather than simply a single-species type of mammal, analysis of benefits and risks needs to be directed toward the whole human, the superorganism. To enable this, old 20th century scientific dogmas concerning human health and safety as pertains to the immune system, microbes, human development, and safety evaluation must be discarded, and new paradigms established that align with 21st century science [53].

Dietert and Dietert [53] recently detailed seven 20th century dogmas that unduly affect science policy even though they are based on now-disproven science. The authors contend that these outdated scientific dogmas are impeding a progression toward much-needed sustainable healthcare. At least three of these outdated dogmas impact the consideration of benefit–risk for milk microbiomes: (1) the incorrect notion that the newborn's immune system is completely balanced and fully functional at birth (significant immune maturation must happen in the infant to avoid predictable diseases); (2) the idea that all microbes are dangerous (most microbes are safe and many are needed); and (3) the idea that it is sufficient for safety assessment to focus on the human mammal (to the exclusion of safety for the human microbiome). This last outdated dogma resulted in existing approved drugs that present significant health risks for humans as they damage the human microbiome, such as proton pump inhibitors [53,54].

Knowledge about the microbiota of milks is contributing to a dramatic transformation of roles, not just of medical professionals, but also of parents and regulators, as 'microbial managers' of healthy microbiomes, to reduce susceptibility to or prevent disease and actively promote health [23]. This microbial management strategy is consistent with a need to shift emphasis from the epidemiologic disease triangle to a health triangle featuring the microbiota [49], as endorsed by others [55,56].

### 4.3. Updating Preconceived Notions on Breastmilk Ecosystem Structure and Function

Clearly, raw milks are not sterile [1–5,11], nor are the microbes present simply contaminants originating from feces or the environment. Rather, raw breastmilk is said to contain an "inimitable plethora of bioactive factors" that act "synergistically, making it difficult to delineate the specific functions of a given milk component" in isolation from the plethora of other factors present [44]. Despite substantial bodies of evidence linking beneficial effects to raw breastmilk (Figure 1; Supplemental Table S1), plausible mechanisms are incompletely characterized to date, largely due to the complexity of the functional networks of interactions within and between natural microbiota and other bioactive components [30,57,58].

The body of evidence documented herein provides strong support for the milk microbiota as beneficial for offspring development and maturation of GI, immune, neural, and respiratory systems in offspring. Further, evidence supports multiple origins of the microbes present. Recent reviews cite studies providing evidence regarding potential origins (niches, sources) of the natural microbiota of milks [2–5,11]. Thorough discussions

of the evidence for breastmilk microbiota were provided by Zimmerman and Curtis [4] and Boudry and colleagues [11], and Oikonomou and colleagues [3] considered evidence on origins of milk microbiota across mammals.

Multiple lines of evidence support the plausible transfer of microbes from the infant oral (bucchal) and the maternal skin microbiomes, as well as an entero-mammary pathway for transfer of microbes or their DNA from the maternal GI tract to mammary tissue and subsequently to milk and the oral cavity and GI tract of breastfeeding infants [4,11,59]. Zimmerman and Curtis [4] document transfer of these gut bacterial genera to breast-milk: *Bacteroides*, *Bifidobacterium*, *Blautia*, *Clostridium*, *Collinsella*, *Cutibacterium*, *Enterococcus*, *Escherichia*, *Lactobacillus*, *Parabacteroides*, *Pediococcus*, *Staphylococcus*, *Streptococcus*, and *Veillonella*. Consistent with Zimmerman and Curtis, additional reviews [60,61] conclude that the predominance of available scientific evidence supports the entero-mammary pathway of transferring maternal GI microbes to breastmilk and breastfeeding infants. In multiple studies, probiotic strains administered during pregnancy were detected in the breastmilk ecosystem [4]. The review by Oikonomou and colleagues [3] cites some of this evidence, and concludes that the body of evidence suggests transfer of microbes from milk to infants via an entero-mammary route, though mechanistic details are not fully understood [11].

### 4.4. Including Benefits of Microbiota-Mediated Colonization Resistance

Given the natural microbiota of milks described above, competition within and between microbes in breastmilk is likely, and networks of microbes are linked by differential functional activities by direct and indirect competitive and cooperative relationships [57]. Similarly, direct and indirect microbial competition of the breastmilk microbiota with potential enteropathogens provide a primary disease prevention strategy with opportunities for more holistic, ecological approaches to 'optimized colonization resistance' in neonatology [9]. The importance of including evidence on dose- and time-dependent colonization resistance in microbial benefit and risk assessments was emphasized in recent studies [23,49].

Extensive data are now available that characterize plausible mechanisms driving infectious and inflammatory disease, including colonization resistance by direct and indirect competition of the microbiota in foods and the gut (Figure 1; Table 2; Supplemental Table S1). Colonization resistance likely enhances the health of superorganisms by multiple mechanisms simultaneously, including: (i) outcompeting pathogens for resources in the intestinal lumen; (ii) reducing likelihood of pathogen attachment along mucosal surfaces of the gut; (iii) up-shifting pathogen load requirements for disease (enhancing innate resistance against low pathogen doses); (iv) strengthening mucosal barriers against pathogenesis; and (v) optimizing immune homeostasis, balancing inflammatory processes linked with allergies, asthma, and infectious disease [9,62].

**Table 2.** Plausible mechanisms for colonization resistance (derived from Kim et al. [62]; Dietert [9]).

| Direct Mechanisms of Microbiota-Medicated Colonization Resistance | Indirect Mechanisms of Microbiota-Medicated Colonization Resistance: |
|---|---|
| **Outcompete enteropathogens for:**<br>• Nutrients, vitamins, co-factors, and minerals;<br>• Niches (function of species in a habitat or environment; interactions of species and abiotic factors of a habitat which supports a community;<br>• Attachment or invasion sites (e.g., mucin, target host cell receptors). | • Stimulating innate immune cells via receptors;<br>• Stimulating host cells to produce antimicrobial compounds (e.g., defensins);<br>• Enhancing the toxicity of bile acids for pathogens;<br>• Activating host immune cells to more effectively clear pathogens (e.g., inflammasome pathway activation);<br>• Maintaining and protecting the mucin layer and underlying gut barrier integrity from damage by pathogens. |
| **Antagonize or kill enteropathogens by directly producing:**<br>• Antimicrobials (e.g., bacteriocins such as Nisin and Plantaricin 423);<br>• Type IV secretion compounds (e.g., VirB/D4 proteins). | |

Application of next generation microbial ecology and combinations of in vitro, in vivo, and microcosm experiments may support evidence-based policies for donor breastmilk that more effectively establishes a healthy gut microbiota and restores colonization resistance against nosocomial [63,64] and food-borne pathogens [62,65,66] in NICU infants in the future.

### 4.5. Updating Earlier Notions from Decision Science

The assumption that pasteurized donor milks are more beneficial and less risky to NICU infants is not supported by definitive evidence, particularly from microbiologic and immunologic perspectives (Figure 1). One could argue that the body of evidence is consistent with decreased health benefits with pasteurization, with the recent exception of a study documenting reduced risk for short-term pasteurization and CMV rates [30]. For some clinical outcomes, pasteurization may actually increase health risks to infants as evidenced in multiple clinical studies [6–8,28,29].

For nearly 80 years, Norway has documented no adverse effects or extra risk associated with its policy of screening and delivering raw breastmilk with non-detectable pathogens to mothers who are unable to breastfeed their infants, and pathogen-positive donor milk is destroyed rather than pasteurized. Including the full body of evidence, both supporting and attenuating, is important to expand deliberations around the world. Further, future deliberations should include evidence for potential pathogens representing high risk to infant health that merit screening in donor milk, not only to protect infant health, but also to maximize benefits to infant health.

Decisions by human donor milk banks to pasteurize all donor milk suggest that implicit barriers are preventing these organizations from considering the 21st century evidence for the breastmilk microbiota that might inform future evidence-based policies that account for both benefits and risks. It is possible that one of these implicit barriers is belief in the 20th century paradigm of germ theory based on fear that presence of potential pathogens alone causes high risk. The needs for deliberation of the evidence and analysis of both benefits and risks are urgent to optimize benefits and risks to NICU infants.

At present, the loss of the well-established benefits for raw breastmilk for many significant endpoints in neonatology by pasteurization [6–8,28,29] appears inconsistent with the 'state of the science' and uncertainties for infant health, and pasteurization appears associated with increased risk to NICU infants around the world. International deliberation of the evidence documented in Figure 1 is essential to development of evidence-based policies. The more holistic 21st century perspectives of the complexity and resilience of raw milk ecosystems documented herein and healthy gut ecosystems [23] may well benefit human health into childhood and adulthood, and outweigh acute infectious disease risks to NICU infants.

The authors propose the evidence map for the breastmilk ecosystem (Figure 1) as a starting point for an international workshop on the benefits and risks of pasteurization. Raw breastmilk and other foods containing a natural microbiota appear to contribute to GI, immune, neural, and respiratory system health for infants and adults. The workshop could launch a series of exercises of an analytic-deliberative process [67,68] to build shared understanding between experts and stakeholders in support of evidence-based risk management decisions on donor milk and other foods.

## 5. Future Direction

This analysis reflects the explosions of knowledge of the microbiota of milks and their functions in health as the 'microbiome revolution' continues to transform into 'normal science', in Kuhn's terminology [48]. North [69] also describes a 'causal revolution' that motivates change in analytical thought, particularly regarding causality in science. Tremendous advances in the state of knowledge about milk microbiota are highlighted from the past decade, from microbiologic, immunologic, and decision science perspectives. Of particular importance for future deliberations regarding evidence for benefits and risks

is the strength of evidence for making valid statistical inferences about balancing benefits and risks, while acknowledging uncertainties about likelihood and magnitude for both.

Importantly, much of the evidence for milk risks is correlative, not causal. For example, just as children's shoe size is correlated to reading ability, but is not a causal predictor [70,71], the potential pathogens present in breastmilk may not cause disease. The danger of managing risks while ignorant of confounded causal factors for benefits and risks is that interventions (e.g., pasteurization) may not reduce risk as intended, but may actually increase risk and reduce benefits [69]. Societal barriers to maximizing health benefits for NICU infants may also include difficulties in shifting paradigms for 'entrenched bureaucratic practice and a highly successful advertising campaign supporting the old paradigm' [68].

Further, urgent needs for greater transparency with stakeholders about the available data, more robust analysis using appropriate methods of inferencing, and exercise of analytic-deliberative process [68], as noted in the quotation below, also apply to future deliberations about evidence for the raw breastmilk ecosystem.

> 'The biggest challenge, generalizing from the phrases cited earlier from the Orange Book [67], is to formulate the right analysis for the problem, and then, working with scientists and other experts who have the relevant information and experience, and the stakeholders who are concerned with the consequences, to do the analysis right and communicate it so the insights from the analysis are understood.'

As discussed previously, outdated dogma from 20th century science is an inappropriate basis for policies and regulations for raw breastmilk that is clearly associated with both benefits and risks. Wider deliberation of the evidence regarding the complexities of the breastmilk ecosystem is needed to develop shared understanding of recent scientific advances before the application of formal qualitative or quantitative microbial benefit–risk analysis for breastmilk. Until such efforts are undertaken, outdated notions about the microbiota of milks will deter development of evidence-based policies for raw and pasteurized milks. The evidence map generated in this study is anticipated to form the basis of an international workshop that will address the changing paradigm of breastmilk ecosystems and initiate the first cycle of analysis and deliberation with decision makers and stakeholders in the 21st century.

The future task of quantitating and balancing the benefits and risks of milk pasteurization is quite demanding, as the body of scientific evidence is extensive, confounded, ambiguous, and open to alternative interpretations. Additionally, relationships between scientific evidence and risk assessment results are nuanced and conditional upon assumptions and limitations of knowledge of the analysts at the time [39]. Further, advocates of any worldview, including opinions about benefits and risks of pasteurization for breastmilk, can be subject to confirmation bias, selectively citing studies that agree with their ideology and worldview, and excluding or dismissing others that provide conflicting evidence. Families are rightly confused by such strong conflicting opinions that exclude evidence that could directly impact the health and survival of their NICU infants. In our view, the microbiology and microbial risk analysis communities could contribute to future deliberations and analyses that bridge the divide for a prominent worldview limiting access to raw breastmilk in NICUs around the world.

**Supplementary Materials:** The following are available online at https://www.mdpi.com/article/10.3390/applmicrobiol1030027/s1. Table S1a: Supplemental Studies on Plausible Mechanisms and Relevance for Pro-Argument on Breastmilk Benefits (Figure 1), Table S1b: Supplemental Studies on Plausible Mechanisms and Relevance for Contra-Argument on Breastmilk Risks (Figure 1).

**Author Contributions:** Conceptualization, M.E.C., R.R.D. and D.W.N.; methodology, M.E.C.; formal analysis, M.E.C.; investigation, M.E.C. and R.R.D.; resources, M.E.C.; writing—original draft preparation, M.E.C.; writing—review and editing, M.E.C., R.R.D. and D.W.N.; visualization, M.E.C. and

M.M.S.; project administration, M.E.C.; funding acquisition, M.E.C. All authors have read and agreed to the published version of the manuscript.

**Funding:** Largely unfunded project with partial support from crowdfunding.

**Institutional Review Board Statement:** Not applicable.

**Informed Consent Statement:** Not applicable.

**Data Availability Statement:** Not applicable.

**Acknowledgments:** The authors appreciate helpful comments from journal reviewers, especially the reviewer who suggested additional studies for inclusion. We are grateful to those individuals and organizations who contributed to a crowdfunding campaign that partially funded this work. We acknowledge helpful comments from Joanne Whitehead, Richard Williams, and Amy Vasquez and Janice Dietert on a previous version of this work. We are also grateful to many members of the Society for Risk Analysis who contributed to discussions of the evidence for this project in recent years.

**Conflicts of Interest:** Author have no potential conflict of interest to declare except R.R.D. who has consulted for a probiotics company on education.

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
