# Peer review of "Examining Evidence of Benefits and Risks for Pasteurizing Donor Breastmilk"

_2673-8007, doi:10.3390/applmicrobiol1030027_

Round 1

Reviewer 1 Report

The manuscript is a timely update on an area of growing clinical interest and controversy. The authors are clearly strong advocates for reconsideration of the need for pasteurization of donated breastmilk.  As such, the submitted manuscript overt emphasizes potential advantages of avoiding pasteurization and minimizes of the real and theoretical risks of using unpasteurized breastmilk. As appropriate for a microbiology review, there is also a lack of exploration of other consequences of pasteurization of donor milk, including an alteration in hormone content, and perhaps the title should be modified in that regard. If the focus is to remain on microbiological risks and benefits, some discussion should be provided for the option of pasteurizing donor milk to reduce clearly or potentially pathogenic bacteria, then providing potentially beneficial bacteria as a probiotic supplement. This option would also improve the consistently of the provided donor milk microbial composition that would otherwise vary from donor to donor independent of the status of the receiving maternal-infant dyad. Operationally, it is also unclear how sequentially using milk from multiple donors with different microbiomes over the course of breast feeding would impact the longitudinal microbiome and thus health of the recipient infant. Beyond those higher-level considerations, the following specific points are offered for consideration.

  1. Repetitive word choice to drive-home bullet points is distracting and unnecessary, e.g., “dense and diverse” being used thrice in the second paragraph and at least 11 times overall.
  2. Introductory paragraph 3 seems to conclude that pasteurized donor milk is a harmful feeding practice, perhaps that is a hypothesis or else should be supported by evidence.
  3. The 6th paragraph ends with implication that some entities value economics over health and only cites an unpublished manuscript. That sentence appears unnecessary conjecture.
  4. The described literature search is not systematic and should at least detail the timing and results of the most recent comprehensive review of all available literature, not just aggressive pursuit of literature and problematic personal communications that are uniformly supportive of the use of raw donor milk.
  5. Page 7, Under the subheading “Attenuating” there are a series of incomplete sentences of unclear relevance that should be improved or moved to a properly formatted section or table. The section’s final sentence that begins “Anti-tumor properties” should be referenced or removed.
  6. Even if risk of HIV transmission via breastmilk is extremely low, there is likely an even lower level of risk tolerance among a vast majority of the general population, and studies that demonstrate such risk and an elimination of risk with pasteurization support rather than attenuate the contra-argument.
  7. A similar argument could be made for use of non-pasteurized CMV positive milk in preterm infants. While human blood is not pasteurized, it is also collected in a sterile fashion and then leukoreduced to decrease CMV transmission. The quotes attributed to Dr. Nordeng are not appropriate.
  8. Is there a “Figure 2” or do the authors mean “Figure 1” in section 3.1.3)?
  9. Discussion, second paragraph, second sentence ignores the fact that certain pathogens in breast milk would certainly lead to discarding or at least pasteurizing milk
  10. Repetitive labeling of current milk bank practices that the authors’ disagree with as “20th century” is distracting and counter-productive to an attempt to build support or consensus.

Author Response

Responses to Reviewer Comments for

Examining Evidence of Benefits and Risks for Pasteurizing Donor Breastmilk

Friday, September 3, 2021

Reviewer 1:  The manuscript is a timely update on an area of growing clinical interest and controversy. The authors are clearly strong advocates for reconsideration of the need for pasteurization of donated breastmilk.  As such, the submitted manuscript overt emphasizes potential advantages of avoiding pasteurization and minimizes of the real and theoretical risks of using unpasteurized breastmilk.

Thank you for these comments. While the authors respectfully disagree with our characterization as advocates, we are concerned about this possible perception. As a result, we have reworded multiple sections of the manuscript to address this possible misperception.

We prepared a major revision marked with redline/strikeout changes in response to comments from both reviewers, beginning with modifications of the tone and content of the Introduction, as well as revisions of the Abstract and some Discussion sub-sections. Specific changes are highlighted in subsequent responses.

From our perspective, we agree with Reviewer 1 that we offer a unique viewpoint of the available body of evidence as risk analysts. We applied established methodology to generate a structured evidence map as a necessary step to fostering open, trans-disciplinary deliberations about benefits and risks of pasteurizing donor milk. We did not estimate benefits or risks, or weigh them; we merely structured the available evidence with an emphasis on recent clinical studies and plausible mechanisms. Remaining uncertainties are communicated to motivate deliberation and additional research and analysis, as well as to inform future evidence based policies as communicated in the new closing sentence of the revised Abstract and a new sentence in section 4.0 regarding the long-term aims of the evidence map to foster deliberation, motivate additional research and analysis, and inform future evidence-based policies about pasteurizing donor breastmilk.

Over the past five years of this project, our searches yielded very extensive evidence of benefits and very sparse evidence of harm for raw breastmilk. We have not found quantitative microbial risk assessment work to support the policy of pasteurization. We would appreciate citations from the reviewer for any studies that are relevant to decisions about pasteurizing donor milk that were overlooked.

While it may appear that we have intentionally overemphasized benefits and minimized risks, from our perspective, our work reflects the nature and extent of the body of evidence, not our opinions about the evidence or policy decisions. We considered how to convey this more clearly and transparently in the revision, starting with the revised Abstract included below.

An evidence map is visualized as a starting point for deliberations by trans-disciplinary stakeholders including microbiologists with interests in the evidence and its influence on health and safety. Available evidence for microbial benefits and risks of the breastmilk ecosystem was structured as an evidence map using established risk analysis methodology. The evidence map based on published literature and reports included the evidence basis, pro- and contra-arguments with supporting and attenuating evidence, supplemental studies on mechanisms, overall conclusions, and remaining uncertainties. The evidence basis for raw breastmilk included one benefit-risk assessment, systematic review, and systematic review/meta-analysis, and two cohort studies. The evidence basis for benefits was clear, convincing, and conclusive, with supplemental studies on plausible mechanisms attributable to biologically active raw breastmilk. Limited evidence was available to assess microbial risks associated with raw breastmilk and pasteurized donor milk. The evidence map provides transparent communication of the ‘state-of-the-science’ and uncertainties for microbial benefits and risks associated with the breastmilk microbiota to assist in deeper deliberations of the evidence with decision makers and stakeholders. The long-term aims of the evidence map are to foster deliberation, motivate additional research and analysis, and inform future evidence-based policies about pasteurizing donor breastmilk.

Additional redline/strikeout changes are noted in the revision and highlighted below.

As appropriate for a microbiology review, there is also a lack of exploration of other consequences of pasteurization of donor milk, including an alteration in hormone content, and perhaps the title should be modified in that regard. If the focus is to remain on microbiological risks and benefits, some discussion should be provided for the option of pasteurizing donor milk to reduce clearly or potentially pathogenic bacteria, then providing potentially beneficial bacteria as a probiotic supplement.

The authors agree that the focus of our work is on microbial benefits and risks. Although we considered adding the word ‘Microbial’ or ‘Microbiological’ before ‘Benefits and Risks’ in the title, we prefer to add text included in the abstract above (and body of the manuscript) that clarifies our scope as microbial benefits and risks.

As to the suggestion of a risk management scenario of reducing pathogens and providing probiotics in donor milk, we believe that this scenario is already addressed appropriately in our companion manuscript, Enhancing Human Superorganism Ecosystem Resilience by Holistically ‘Managing our Microbes’. We agree that this scenario is appealing from emerging knowledge of recent research in microbial ecology and microbial risk analysis. However, we have found little definitive evidence from clinical studies or other research to support the reviewer’s scenario. The ‘state of the science’ appears too immature and theoretical at present to endorse such a strategy when our knowledge of the microbiota of breastmilk and its benefits and risks is incoherent and incomplete. In our perspective, our ability to ‘manage our microbes’ for donor breastmilk, though appealing scientifically, appears too limited for practical applications at present.

This option would also improve the consistently of the provided donor milk microbial composition that would otherwise vary from donor to donor independent of the status of the receiving maternal-infant dyad. Operationally, it is also unclear how sequentially using milk from multiple donors with different microbiomes over the course of breast feeding would impact the longitudinal microbiome and thus health of the recipient infant.

We agree with the reviewer that potential longitudinal effects of multiple donors on microbiome and infant health are not clear. We also agree that such effects for multiple donors, exclusively or in addition to mother’s own milk, on infant health might be an important issue for deliberation.

However, from our perspective, the acute need for daily ingestion of milk microbes in the NICU, whether mother’s own or from a donor or multiple donors, to ‘feed and seed’ a protective gut microbiota might outweigh theoretical long term risks and uncertainties for exposures to milk from multiple sources. We would appreciate citations from the reviewer for any studies that describe the nature and magnitude of longitudinal impacts of multiple donors on infant health if such studies were overlooked.

Beyond those higher-level considerations, the following specific points are offered for consideration.

  1. Repetitive word choice to drive-home bullet points is distracting and unnecessary, e.g., “dense and diverse” being used thrice in the second paragraph and at least 11 times overall.

Thank you for this comment. Our intention was not to distract, but to emphasize the microbial ecology of the breastmilk microbiota, particularly its dense and diverse microbes, as a crucial aspect for healthy growth and development of infants. In our revision, we suggest at first use, defining ‘natural microbiota’ as being ‘dense and diverse’ so that we emphasize the microbial ecology of the system. Thereafter, we use the term ‘natural microbiota’ in the following paragraphs of the revision, indicated by redline.

  • Paragraphs 1, 2, 3 and 4 of Introduction,
  • Paragraph 1 of Discussion
  • Paragraph 1 of Discussion subsection 4.1
  • Paragraphs 1 and 2 of Discussion subsection 4.2
  • Paragraphs1 of Discussion subsection 4.3
  • Last paragraph of Discussion subsection 4.4
  1. Introductory paragraph 3 seems to conclude that pasteurized donor milk is a harmful feeding practice, perhaps that is a hypothesis or else should be supported by evidence.

Thank you. We agree that some of the text is more suited to introduce the Discussion section than to include in the Introduction for this work. The paragraph and some surrounding text was deleted from the Introduction and moved to the beginning of the Discussion section, as noted in the revision.

  1. The 6th paragraph ends with implication that some entities value economics over health and only cites an unpublished manuscript. That sentence appears unnecessary conjecture.

Thank you for this comment. While the implication noted by the reviewer may be true, we acknowledge that this statement does not describe risk assessment. The statement does build on work by the WHO that we cited in the same paragraph. From our perspective, we do not agree that the statement is unnecessary conjecture, but rather view it as relevant for risk analysis, specifically consideration of potential risk management tradeoffs.

Consistent with our revised abstract, our longterm aims are to map the evidence and address uncertainties in a manner amenable to future trans-disciplinary deliberations that will include both health and economic evidence. We revised the sentence to read as follows in the revision.

Regarding risk management for infants fed breastmilk and formula, a companion manuscript also submitted to this special collection (Coleman et al. 2021) addresses relevant issues of ‘managing our microbes’ and raises concerns about potential tradeoffs between economic motivations of the global infant formula industry and health benefits of raw breastmilk to infants and families.

Regarding the reviewer’s concern about citing a submitted by yet unpublished manuscript, we note that we are still awaiting the peer reviews on the companion paper. Also, we do not object if the editorial office for the journal provides Reviewer 1 the companion manuscript as it is relevant to this comment.

  1. The described literature search is not systematic and should at least detail the timing and results of the most recent comprehensive review of all available literature, not just aggressive pursuit of literature and problematic personal communications that are uniformly supportive of the use of raw donor milk.

We agree that the literature search was not ‘systematic’ in the sense of conducting a ‘systematic review’. We agree that our work is not a systematic review at all. We also responded to Review 2 that we do not view our work as a review. We submitted our work as an original research article because we see the evidence map as a unique contribution beyond a literature review.

Agreeing that our work is not a systematic review, we did perform systematic searches that spanned ~5 years, with input from regulators and scientists from around the world. Dates were inserted in the search description of the Methods section, as requested.

We view the evidence map as a comprehensive and objective outcome to serve as the basis for wider deliberation, with other microbiologists and decision scientists, as well as other trans-disciplinary perspectives with a ‘stake’ in decisions about pasteurizing donor milk (stakeholders as commonly described in the risk arena).

We are concerned about the perception of our work as ‘aggressive pursuit of literature and problematic personal communications’.

On the contrary, we carefully searched, reviewed, structured, and cited the available evidence from clinical studies and the risk literature. As stated above, we do not characterize ourselves as advocates. We may appear to overemphasize benefits and minimize risks, but that is the nature and extent of the supporting evidence that the reviewer appears to agree merits wider deliberation.

We would appreciate citations from the reviewer for any studies that the reviewer feels offer evidence on benefits and risks of pasteurizing breastmilk but were overlooked.

We considered how to convey the scope, intent, and purpose of our work more clearly and transparently in the revision. We deleted the quotes from Dr. Nordeng as requested below in point 7. The only personal communication that we cited regarded specific reports of screening donor breastmilk by a major hospital from the first author of a published peer-reviewed manuscript on this subject (Grøvslien, 2020; Grøvslien & Grønn, 2009).

The revised manuscript now centers more strongly on the available scientific evidence and provides a structured map of the evidence on benefits and risks for wider deliberation of the evidence relevant to decisions about pasteurizing donor milk, in our opinion.

  1. Page 7, Under the subheading “Attenuating” there are a series of incomplete sentences of unclear relevance that should be improved or moved to a properly formatted section or table. The section’s final sentence that begins “Anti-tumor properties” should be referenced or removed.

Thank you for these comments.

We considered the text in this sub-section and inserted the following statement in the revision before five numbered bullets of incomplete sentences: ‘These plausible mechanisms include the following 5 examples:’.

Regarding the comment about anti-tumor properties, we revised this sentence and added a reference, as recommended.

  1. Even if risk of HIV transmission via breastmilk is extremely low, there is likely an even lower level of risk tolerance among a vast majority of the general population, and studies that demonstrate such risk and an elimination of risk with pasteurization support rather than attenuate the contra-argument.

Thank you for this comment. We did insert text to clarify acronyms in the first paragraph of this sub-section as indicated in the redline of the revision. Note that this sub-section emphasized the paucity of evidence on risk, with no QMRA studies identified in our searches. We would appreciate citations from the reviewer for any studies that the reviewer feels offer evidence on risks of pasteurizing breastmilk but were overlooked.

Further, we agree that some studies may provide both evidence that supports and attenuates the arguments. This reviewer comment prompted us to expand our explanation (in Section 3.1.2) of the American Academy of Pediatrics position paper (AAP 2017) in the revision, under Supporting studies. This AAP paper and Lindemann et al. (2004) and Keim et al. (2013) cited therein similarly provide both supporting and attenuating evidence that merit deeper deliberation, particularly in light of advances in knowledge of the natural microbiota of breastmilk cited in our work.

  1. A similar argument could be made for use of non-pasteurized CMV positive milk in preterm infants. While human blood is not pasteurized, it is also collected in a sterile fashion and then leukoreduced to decrease CMV transmission. The quotes attributed to Dr. Nordeng are not appropriate.

We can agree to delete the quotes, even though we do believe that Dr. Nordeng’s expert opinion is valid input to decisions about pasteurizing donor milk.

  1. Is there a “Figure 2” or do the authors mean “Figure 1” in section 3.1.3)?

We apologize for the editorial error. We included a single figure for this manuscript, Figure 1. Note that Figure 1 was revised for consistency with the referencing style specified for the journal, now communicating reference numbers.

  1. Discussion, second paragraph, second sentence ignores the fact that certain pathogens in breast milk would certainly lead to discarding or at least pasteurizing milk

We are confused about the factual basis and certainty that the reviewer refers to in this comment. In our perspective, the evidence is far from coherent and factual regarding the predictive value of the presence of potential pathogens in donor milk for estimating likelihood of health and disease outcomes or the benefits and risks of discarding or pasteurizing donor milk.

Further, from the perspective of microbial ecology, medical microbiology, and risk analysis, decisions about donor milk do not appear to be supported by definitive factual evidence for either benefits or risks. Neither are we aware of any studies that estimate benefits and risks, estimate potential effectiveness of risk management scenarios for interventions to reduce risks, or weigh benefits and risks for donor milks. That is the major point of our manuscript, that wider deliberation of the available body of evidence is needed so that this work can be undertaken.

The revision addresses this concern in clarifying the aims of the study, the ‘state of the science’ and uncertainties, and needs for additional research and analysis for risk assessment and risk management scenarios that could be undertaken in the future.

  1. Repetitive labeling of current milk bank practices that the authors’ disagree with as “20th century” is distracting and counter-productive to an attempt to build support or consensus.

Thank you for this comment. We revised the Discussion section extensively to reduce repetition and distraction. We removed terms 20th and 21st century in sub-headings, and added a new sub-section to the discussion on scientific revolutions. We replaced the phrase ‘Updating Preconceived Notions from 20th Century Medicine and Risk Assessment’ in the original Discussion sub-heading 4.1 with ‘Updating Earlier Notions from Science, Medicine, and Risk Analysis’. We retained examples citing 20th and 21st century paradigms that appear to us to be significant, particularly as barriers to deliberation of the available evidence.

20 July 2021                                                                                              Manuscript Submission Date

19, 22 Aug 2021                                                                                                  Dates of these reviews

3 Sep 2021                                                                                                           Date of this response

Reviewer 2 Report

The work of Coleman et al. have conducted a good system review for the important and unsolved issue for the raw or pasteurization of human milk. The figure 1 is beautiful and informative.  The manuscript was well written and fruitful. Several points need to be confirmed.

  1. The type of the article should be considered as a review article.
  2. Please confirm and mark the correspondence author in the author list.
  3. Please apply the right citation style of this journal to the article.
  4. Please clarify the aim of this review for the abstract.
  5. Please confirm the table caption of table 1. Abbreviation in the tables should be listed the footer.
  6. About `2020 communication from University of Oslo, https://partner.sciencenorway.no/health-health-services-medical-procedures/high-standards-for-human-milk-banks-in-norway/1617690)' , the long URL could be listed a citation.

Author Response

Responses to Reviewer Comments for

Examining Evidence of Benefits and Risks for Pasteurizing Donor Breastmilk

Friday, September 3, 2021

Reviewer 2:  The work of Coleman et al. have conducted a good system review for the important and unsolved issue for the raw or pasteurization of human milk. The figure 1 is beautiful and informative.  The manuscript was well written and fruitful. Several points need to be confirmed.

  1. The type of the article should be considered as a review article.

Thank you for the comments. However, we submitted this work as an original research article because we did not undertake it as a systematic review.

We view our work as original and analytical. The Reviewer notes that the figure, an evidence map, is ‘beautiful and informative’. We see that figure, the evidence map, as an output of our original analytical work that structures the evidence as a starting point for future deliberations by trans-disciplinary stakeholders with interests in benefits and risks of pasteurizing donor breastmilk.

Note that Reviewer 1 stated concerns that our work did not meet expectations for a systematic review. We believe that other readers of our work would also raise this concern if we categorized the analysis as a review.

We prefer to continue to list our work as an article rather than as a review to minimize the likelihood of raising concerns that the work was a sloppy systematic review rather than an objective analysis of the available evidence to support wider deliberations.

  1. Please confirm and mark the correspondence author in the author list.

Apologies for the omission. M.E. Coleman is the corresponding author as noted in the revision.

  1. Please apply the right citation style of this journal to the article.

Thank you for the comment. The revision includes the reference style specified for the journal. Note that we also inserted reference numbers in a revised Figure 1 in a hybrid style explained in the figure legend listed below.

Figure 1. Evidence Map for Raw Breastmilk Ecosystem. B-RA = Benefit-Risk Assessment; QMRA = Quantitative Microbial Risk Assessment; CS = human cohort study; MA = meta-analysis; R = review; RT = randomized trial; SR = systematic review. NEC = necrotizing enterocolitis. Referencing for this figure lists first author and reference number in brackets for those references cited in the text or first or first and second author(s) and year for Supplemental Studies on Mechanisms for which full references are provided in Supplemental Appendix 1.

  1. Please clarify the aim of this review for the abstract.

Thank you for the comment. We suggest the following wording for the abstract in the revision that clarifies this point.

An evidence map is visualized as a starting point for deliberations by trans-disciplinary stakeholders including microbiologists with interests in the evidence and its influence on health and safety. Available evidence for microbial benefits and risks of the breastmilk ecosystem was structured as an evidence map using established risk analysis methodology. The evidence map based on published literature and reports included the evidence basis, pro- and contra-arguments with supporting and attenuating evidence, supplemental studies on mechanisms, overall conclusions, and remaining uncertainties. The evidence basis for raw breastmilk included one benefit-risk assessment, systematic review, and systematic review/meta-analysis, and two cohort studies. The evidence basis for benefits was clear, convincing, and conclusive, with supplemental studies on plausible mechanisms attributable to biologically active raw breastmilk. Limited evidence was available to assess microbial risks associated with raw breastmilk and pasteurized donor milk. The evidence map provides transparent communication of the ‘state-of-the-science’ and uncertainties for microbial benefits and risks associated with the breastmilk microbiota to assist in deeper deliberations of the evidence with decision makers and stakeholders. The long-term aims of the evidence map are to foster deliberation, motivate additional research and analysis, and inform future evidence-based policies about pasteurizing donor breastmilk.

  1. Please confirm the table caption of table 1. Abbreviation in the tables should be listed the footer.

Thank you for the comment. We added a table title consistent with the journal style.

  1. About `2020 communication from University of Oslo, https://partner.sciencenorway.no/health-health-services-medical-procedures/high-standards-for-human-milk-banks-in-norway/1617690)' , the long URL could be listed a citation.

Thank you for the suggestion. However, we decided to delete these quotes as per Point 7 from Reviewer 1.

20 July 2021                                                                                              Manuscript Submission Date

19, 22 Aug 2021                                                                                                  Dates of these reviews

3 Sep 2021                                                                                                           Date of this response

Round 2

Reviewer 1 Report

The manuscript, especially the abstract, is better balanced with the extensive revisions. As requested, a sampling of citations that offer evidence supporting the pasteurization of donated breastmilk are provided:

  1. Bapistella et al “Short-term Pasteurization of Breast Milk to Prevent Postnatal Cytomegalovirus Transmission in Very Preterm Infants”
  2. Gayà et al. “Analysis of Thermal Sensitivity of Human Cytomegalovirus Assayed in the Conventional Conditions of a Human Milk Bank.”
  3. Novak et al. “Contamination of expressed human breast milk with an epidemic multiresistant Staphylococcus aureus clone”
  4. Ferreira et al. “Pasteurization of human milk to prevent transmission of Chagas disease”.
  5. Eglin RP,Wilkinson AR. “HIV infection and pasteurization of breast milk.”

Regarding the supplemental table, the row for Murphy et al does not appear to be cited in the references and citation 13 is mistitled in the table itself. Many of the citations are problematic, starting with the first that does not specify the reason an entire cohort (those that were to have received L. reuteri) is listed in clinicaltrails.gov but not the manuscript itself.

Author Response

Responses to Review 1 Comments, Round 2, for Examining Evidence of Benefits and Risks for Pasteurizing Donor Breastmilk

Comments and Suggestions for Authors

The manuscript, especially the abstract, is better balanced with the extensive revisions.

Thank you for this comment on improvements in the revision. We began work on the previous revision by accepting all redline/strikeout changes so that the current revision only includes modifications in response to the second round of comments from Reviewer 1.

As requested, a sampling of citations that offer evidence supporting the pasteurization of donated breastmilk are provided:

  1. Bapistella et al. “Short-term Pasteurization of Breast Milk to Prevent Postnatal Cytomegalovirus Transmission in Very Preterm Infants”
  2. Gayà et al.. “Analysis of Thermal Sensitivity of Human Cytomegalovirus Assayed in the Conventional Conditions of a Human Milk Bank.”
  3. Novak et al.. “Contamination of expressed human breast milk with an epidemic multiresistant Staphylococcus aureus clone”
  4. Ferreira et al.. “Pasteurization of human milk to prevent transmission of Chagas disease”.
  5. Eglin RP,Wilkinson AR. . “HIV infection and pasteurization of breast milk.”

Thank you for providing these 5 studies for our consideration.

The two most recent studies (Bapistella et al., 2019; Gayà et al. 2021) were added to this revision. The Bapistella cohort study [new reference 34] is cited in Figure 1, supporting the Contra-Argument, and in the Results sub-section 3.1.2 (page 8 of the revision) and Discussion sub section 4.5 (page 15 of the revision). The Gayà study on thermal treatments to reduce CMV [new reference 40] is cited in the Discussion (pg. 10).

Also in light of these two newly cited studies, we revised the second bullet for the Conclusions text box and the third question for the Remaining Uncertainties text box in the figure, as well as text for Results (Conclusions, Remaining Uncertainties) in redline/strikeout.

Since our evidence map emphasized studies published from 2010 to 2021, we did not add text or cite references 3-5 above that were published from 1987 to 2001. However, we acknowledge that these historical studies and the agents discussed are important for future deliberations. For consistency, in this revision, we added text in Results sub-section 3.1.2 citing additional details from the three more recent studies that describe some of the deeper history for raw breastmilk hazards (Gribble and Hausman, 2012; Keim et al., 2013; American Academy of Pediatrics, 2017). We note that we did not undertake a broad systematic review of evidence from prior decades, but focused our analysis on recent clinical studies, systematic reviews and other analyses from 2010 to present. From our perspective, the results and conclusions would not be changed by additional text citing these older studies, whereas the 2019 and 2021 studies did indeed strengthen the manuscript. Your insights are greatly appreciated.

Regarding the supplemental table, the row for Murphy et al does not appear to be cited in the references and citation 13 is mistitled in the table itself. Many of the citations are problematic, starting with the first that does not specify the reason an entire cohort (those that were to have received L. reuteri) is listed in clinicaltrails.gov but not the manuscript itself.

We appreciate these comments on the Supplemental Table. As we corrected the errors that you identified, we also found other improvements for the Supplemental Tables 1a and 1b. For clarity, we did not use redline/strikeout, as rearrangements would be harder to follow with tracked changes. The revised Supplemental Table now includes Murphy et al., 2017 in reference list (initial revision incorrectly cited [5], Dietert, 2018) and now lists the correct title for Van Den Elsen 2019 (initial revision incorrectly listed title in [13] for Van Daele et al., 2019).

The intent of this table based on Figure 1 is to provide more information to readers with interests in the body of evidence for mechanisms by which benefits and risks of the breastmilk microbiota appear to function. We added rows to the Table for each section heading listed in the text boxes for the supplemental studies of Figure 1. Both Figure 1 and the Supplemental Table lists studies in increasing chronological order (most recent studies listed last). From our perspective, this organization is more consistent and more informative for readers than the previous alphabetic lists of studies in the Supplemental Table. In addition, we added redline/strikeout text in the body of the Method section (paragraph 2) of the revision to indicate what evidence is presented in the Pro- and Contra- Arguments versus in Supplemental Table 1a,b.

Regarding the study that was cited first in the initial Supplementary Table as submitted (Arroyo et al., 2010), this cohort study was designed to compare probiotic and antibiotic treatments of women with mastitis. It does not actually provide evidence on benefits and risk of raw or pasteurized breastmilk ingested by infants. Thus, we did not include it in the Pro- or Contra-Argument sections of the figure or in the body of the manuscript. Rather, the Arroyo study does provide supplemental evidence on plausible mechanisms by which probiotics may function to benefit infants and their mothers, grouped in Figure 1 with other studies, including another study that administered probiotics (Fernandez et al., 2016), related to mechanisms relevant to the microbial ecology of breastmilk.

Thank you again for pointing out issues in the revision that required further clarification. We will acknowledge your helpful comments in the final manuscript.

20 July 2021                                                                                                                 Manuscript Submission Date

19- 22 Aug 2021                                                                                                              Dates of 1st Round Review

3 Sep 2021                                                                                                                    Date of Response to Round 1

9 Sep 2021                                                                                                                   Date of Response to Round 2

Round 3

Reviewer 1 Report

I appreciate the responses and have no new concern.